# Burden of leprosy and associated risk factors for disabilities in Tanzania from 2017 to 2020

George Mrema[1,2]*, Ally Hussein[1,2], Welema Magoge[3], Vida Mmbaga[3], Azma Simba[3], Robert Balama[4], Emmanuel Nkiligi[4], Paul Shunda[4], Deus Kamara[4], Riziki Kisonga[4], Gideon Kwesigabo[1]

**1** Department of Epidemiology and Biostatistics, Muhimbili University of Health and Allied Sciences, Dar es Salaam, Tanzania, **2** Tanzania Field Epidemiology and Laboratory Training Program, Ministry of Health, Dar es Salaam, Tanzania, **3** Epidemiology and Disease Control Section, Ministry of Health, Dodoma, Tanzania, **4** National Tuberculosis and Leprosy Program, Ministry of Health, Dodoma, Tanzania

* drgeorgemrema@gmail.com

## Abstract

### Background

Leprosy is caused by *Mycobacterium leprae* which affects skin, nerves, eyes, and nasal mucosa. Despite global elimination efforts, Tanzania remains among 13 countries reporting more than 1000 leprosy cases annually. In 2021, Tanzania identified 1,511 new cases, with 10% having grade II disability. Moreover, 14 councils recorded leprosy rates exceeding 10 cases per 100,000 population. This study aimed to assess the burden of leprosy and associated risk factors for disabilities in Tanzania from 2017 to 2020.

### Methodology

A retrospective cross-sectional study was conducted to investigate all registered treated leprosy patients from January 2017 to December 2020. The Leprosy Burden Score (LBS) was used to assess the disease burden, while binary logistic regression was employed to evaluate the risk factors for disability.

### Result

A total of 6,963 leprosy cases were identified from 2017 to 2020. During this period, the point prevalence of leprosy declined from 0.32 to 0.25 per 10,000 people, and the new case detection rate decreased from 3.1 to 2.4 per 100,000 people; however, these changes were not statistically significant (p > 0.05). Independent risk factors for leprosy-related disabilities included male sex (Adjusted Odds Ratio (AOR) = 1.38, 95% Confidence Interval (CI) 1.22–1.57), age 15 years and above (AOR = 2.42, 95% CI 1.60–3.67), previous treatment history (AOR = 2.18, 95% CI 1.69–2.82), and positive Human Immunodeficiency Virus (HIV) status (AOR = 1.60, 95% CI 1.11–2.30).

### Conclusion

This study identified male sex, older age, positive HIV status, and prior treatment history as independent risk factors for leprosy-related disabilities. Additionally, despite the observed

**Funding:** The author(s) received no specific funding for this work

**Competing interests:** The authors have declared that no competing interests exist

decline in point prevalence and new case detection rates, these changes were not statistically significant. To address leprosy-related disabilities, it is crucial to implement specific prevention strategies that focus on high-risk groups. This can be accomplished by enhancing screening and contact tracing efforts for early patient identification to prevent delays in intervention. Further research is warranted to analyze the burden of leprosy over a more extended period and to explore additional risk factors not covered in this study.

## Introduction

Leprosy is a chronic infectious disease caused by the slow growing bacteria known as *Mycobacterium leprae*, primarily affecting the skin, peripheral nerves, eyes, and nasal mucosa [1, 2]. Transmission occurs through infectious nasal droplets and prolonged skin contact with untreated patients [3]. Late diagnosis and partial treatment can lead to permanent disability and complications including deformities, amyotrophy, sensory loss, wounds on the hands and feets. These complications often result in social stigma, discrimination, mental health issues, and economic ramifications [4–6].

Leprosy elimination was declared globally in the year 2000, and significant declines in cases were noted until 2005 [7, 8]. Tanzania achieved leprosy elimination status in 2006 [9]. However, despite this progress, the World Health Organization's (WHO) global epidemiological update of 2021 indicated that 13 countries, including Tanzania, still report over 1,000 new leprosy cases annually [10]. In 2021, there were 140,594 new cases worldwide, with 8,492 (6.0%) classified as grade II disability and 9,052 (6.4%) being children under 15 years of age. Regionally, South East Asia reported the highest incidence with 93,485 cases, followed by Africa with 21,201 cases and the Americas with 19,826 cases [10].

In Tanzania, a total of 1,511 new cases were identified in the year 2021, with 154 (10%) presenting with grade II disability and 41 (3%) being children under the age of 15 [10, 11]. The majority of new cases have emerged from 14 councils, each reporting more than 10 cases per 100,000 population [11]. Factors contributing to the high burden of leprosy in these districts include remote locations limiting access to services, the presence of special leprosy camps, and proximity to bordering countries facilitating case importation. Furthermore, funding for leprosy interventions has significantly decreased following the declaration of elimination.

Without innovative strategies in these high-burden districts, leprosy transmission will likely persist, making the disease endemic [12]. Tanzania has implemented various efforts to accelerate leprosy elimination, including a Post-Exposure Prophylaxis (PEP) trial that screened over 4,500 individuals and identified 30 new cases, demonstrating efficacy in early detection. Evaluations of the Bangkok Declaration Special Fund indicate a prevalence rate of less than 1 in 10,000. Despite these advancements, challenges remain, as evidenced by the continued proportion of grade II disabilities and a notable number of affected children. Positive indications from the Country Model Zero Leprosy Review, such as decreasing prevalence rates, are often countered by obstacles like inadequate funding and limitations in contact tracing [11].

To realize leprosy elimination in all localities, strategies must encompass active case searching, contact tracing, and the management of cases and associated complications to halt transmission. Additionally, understanding the trends in leprosy hotspots and the factors associated with disabilities is crucial. This study aims to describe the burden and trends of leprosy, as well as the risk factors for disabilities among cases detected in Tanzania from 2017 to 2020. The findings will serve as a call to action for national and local authorities to enhance Leprosy

Elimination Campaigns (LEC), active case finding, Leprosy Post Exposure Prophylaxis (LPEP), and the therapeutic management for the Prevention of Disability (POD), including rehabilitation services.

## Methodology

### Study design

A retrospective cross-sectional study was conducted among leprosy cases who were enrolled in treatment from January 2017 to December 2020.

### Study area

This study included all 26 regions of Tanzania's mainland and two islands of Zanzibar. In Tanzania, 4,013 out of 8,446 total health facilities function as Multidrug Therapy (MDT) centers, managing leprosy patients and ensuring medication adherence [9].

### Study population

The study population comprised all leprosy cases who were enrolled in treatment from January 2017 to December 2020 as recorded in the National Tuberculosis and Leprosy Program database.

### Variables

The dependent variable in this study was leprosy disability. This was a binary outcome, with the first outcome involving patients with grade 0 disability and the second outcome involving patients with grade I and II disabilities.

Independent variables included age, sex, classification of leprosy, patient category and HIV status.

### Dataset description and data abstraction procedure

Leprosy data collection begins when patients visit health facilities presenting symptoms such as skin patches, numbness, weakness, or ulcers. Upon paying for registration, they see a clinician for diagnosis.

Clinicians diagnose leprosy based on cardinal signs, including hypo-pigmented or reddish lesions with loss of skin sensation, enlarged peripheral nerve trunks with muscle weakness, and the presence of acid-fast bacilli in a slit skin smear or microscopy [13]. If diagnosed, patients are linked to a tuberculosis and leprosy clinic, where trained nurses initiate treatment using the Directly Observed Treatment (DOT) strategy to monitor medication adherence (i.e., a strategy designed to ensure treatment adherence by having a nominated person observe the patient taking the medicine, with a record of each instance of medication intake). Additionally, Patients are assessed for disability and graded as follows: Grade 0 disability means absence of disability (no anesthesia) and no visible damage or deformities on eyes, hands and feet. Grade I disability is when the patient is presenting with loss of protective sensibility in the eyes, hands or feet, but no visible damage or deformities whereas grade II disability is the presence of deformities or visible damage to the eyes (lagophthalmos and/or ectropion, trichiasis, corneal opacity, visual acuity less than 0.1 or difficulty counting fingers at 6 meters), visible damage on hands or feet (hand with ulcerations and/or traumatic, resorption, claw, fallen hand, ulcers; feet with trophic and/or traumatic injuries, resorption, claw, foot drop, ulcers, ankle contracture) [13].

According to Tanzanian policy, leprosy treatment is provided free of charge with treatment duration of6 to 12 months.

After diagnosis, patients are recorded in a facility leprosy register, and their information is forwarded to the District Tuberculosis and Leprosy Coordinator (DTLC), where they are registered in the Electronic Tuberculosis and Leprosy (ETL) System—a web-based surveillance tool for monitoring various national program indicators. This system tracks treatment progress for TB, DR TB, and leprosy patients. Immediately after registration of patients in the system, this information is reflected instantly at all levels (National and Subnational level). Since all patients diagnosed with leprosy are registered in the database, hence these data approximate incidence of leprosy in the country. Nevertheless, there is a possibility of small proportion of missed cases not attending the health facilities for treatment.

Data was extracted from the ETL system in form of MS Excel.

## Data analysis

Data extracted from ETL in form of MS excel was then cleaned and analyzed using Stata version 15.0. Demographic and clinical variables were presented using frequencies and proportions. The regional leprosy burden in the country was determined using LBS developed by WHO [12]. Table 1 below narrates each indicator assessed and score guide.

LBS compose 9 leprosy elimination indicators [12]. High = Light yellow (score = 2), Medium = Gray (score = 1), and Low = Light blue (score = 0) were the three grades assigned to each indicator. The LBS was graded into three levels based on the sum of the individual indicator scores: High = Light yellow (Score 2) when the LBS was equal to or greater than 5, Median = Gray (Score = 1) when the LBS was between 3 and 4, and Low = Light blue (Score-0) when the LBS was 2 and below. Yearly projected population for respective years for calculation of different indicators was obtained from Tanzania National Bureau of Statistics (NBS). LBS was determined for all 26 regions in Tanzania mainland and two islands of Zanzibar from the year 2017 to 2020 and were categorized as high, medium and low.

A map showing regional distribution of leprosy cases from 2017 to 2020 was created using Quantum Geographic Information System (QGIS) software 3.26.3. In addition, maps were drawn to show regional burden of leprosy in the years 2017 and 2020. The map utilized openly available shapefiles (https://www.nbs.go.tz/statistics/topic/gis) obtained from the 2012 population and housing census, which were later modified to accurately depict the information for each region.

To ascertain the statistical significance of a trend of leprosy elimination indicators, chi square test for trend was performed and p-value for respective indicators assessed was reported.

In determining risk factors associated with leprosy disability, binary logistic regression was utilized. Factors such as sex, age group, patient category, classification of leprosy patients and HIV status were all assessed. Factors that were statistically significant at a p-value of ≤0.2 in

**Table 1. Scale for the leprosy burden score (LBS) assessment.**

| Scale | Detection (New cases) for regions | Point prevalence rate per 10,000 population | New Case Detection Rate per 100,000 population | % MB in new Cases | % Children in new cases | % Female in new cases | % Grade II disability in new cases | Grade II disability rate per 100,000 population | [a]P/D ratio | LBS |
|---|---|---|---|---|---|---|---|---|---|---|
| High | >100 = 2 | >2 = 2 | >20 = 2 | <50 = 2 | >20 = 2 | <40 = 2 | >20 = 2 | >1 = 2 | >2 = 2 | ≥5 = 2 |
| Medium | 21–100 = 1 | 1–2 = 1 | 10–20 = 1 | 50–75 = 1 | 10–20 = 1 | >60 = 1 | 10–20 = 1 | 0.5–1 = 1 | 1–2 = 1 | 3–4 = 1 |
| Low | 0–20 = 0 | <1 = 0 | <10 = 0 | 76–100 = 0 | <10 = 0 | 40–60 = 0 | <10 = 0 | <0.5 = 0 | <1 = 0 | 0–2 = 0 |

[a]P/D: Prevalence/Detection

the bivariate analysis were considered as potential risk factors and included in the multivariable model. Adjusted Odds ratio and their respective 95% confidence intervals were reported.

## Ethical clearance

The study reviewed secondary data that are part of routine medical and public health surveillance; therefore, formal ethical clearance was not required. Approval to use the data from the National Tuberculosis and Leprosy Program was obtained from the Permanent Secretary at the Ministry of Health. Confidentiality of patient data was assured by coding the personal identifiers of patients.

## Result

### Demographic and clinical characteristics of leprosy cases

A total of 6963 leprosy cases were notified between the year 2017 to 2020. The disease was found to be common among male patients 4478 (64.3%). Most reported cases 2,061 (38.7%), were between the age of 35 to 54 years. Majority of the notified cases were new patients 6,661 (95.7%) and were classified as having Multi bacillary leprosy (MB) 6,465 (92.9%). Moreover, assessment conducted before treatment initiation have shown that only 655 (9.4%) of the patients had disability grade II. Additionally, regional distribution of leprosy cases showed that majority of cases were reported in Morogoro, Tanga and Dar es Salaam regions. Table 2 and Fig 1 provide detailed information.

### Regional burden of leprosy in Tanzania from the year 2017 to 2020

Over the years 2017 to 2020, the country observed a decline in the point prevalence of leprosy from 0.32 to 0.25 per 10,000 people. Although there was a 21.9% reduction, it was not statistically significant (P = 0.484). Similarly, the New Case Detection Rate (NCDR) showed a decrease from 3.1/100,000 to 2.4/100,000, but the changes were not statistically significant (P = 0.488).

**Table 2. Demographic and clinical characteristics of leprosy patients in Tanzania from 2017 to 2020 (n = 6963).**

| Variable | Frequency (n) | Percentage (%) |
|---|---|---|
| **Gender** | | |
| Female | 2,485 | 35.7 |
| Male | 4,478 | 64.3 |
| **Age group (years)** | | |
| $\leq$14 | 276 | 4.0 |
| $\geq$15 | 6,687 | 96.0 |
| **Category of Patient** | | |
| New | 6,661 | 95.7 |
| Previous treated | 302 | 4.3 |
| **Classification** | | |
| Paucibacillary (PB) | 498 | 7.1 |
| Multibacillary (MB) | 6,465 | 92.9 |
| **Disability grading at the start of treatment** | | |
| Grade 0 | 4,337 | 62.3 |
| Grade I | 1,310 | 18.8 |
| Grade II | 655 | 9.4 |
| Not classified | 661 | 9.5 |

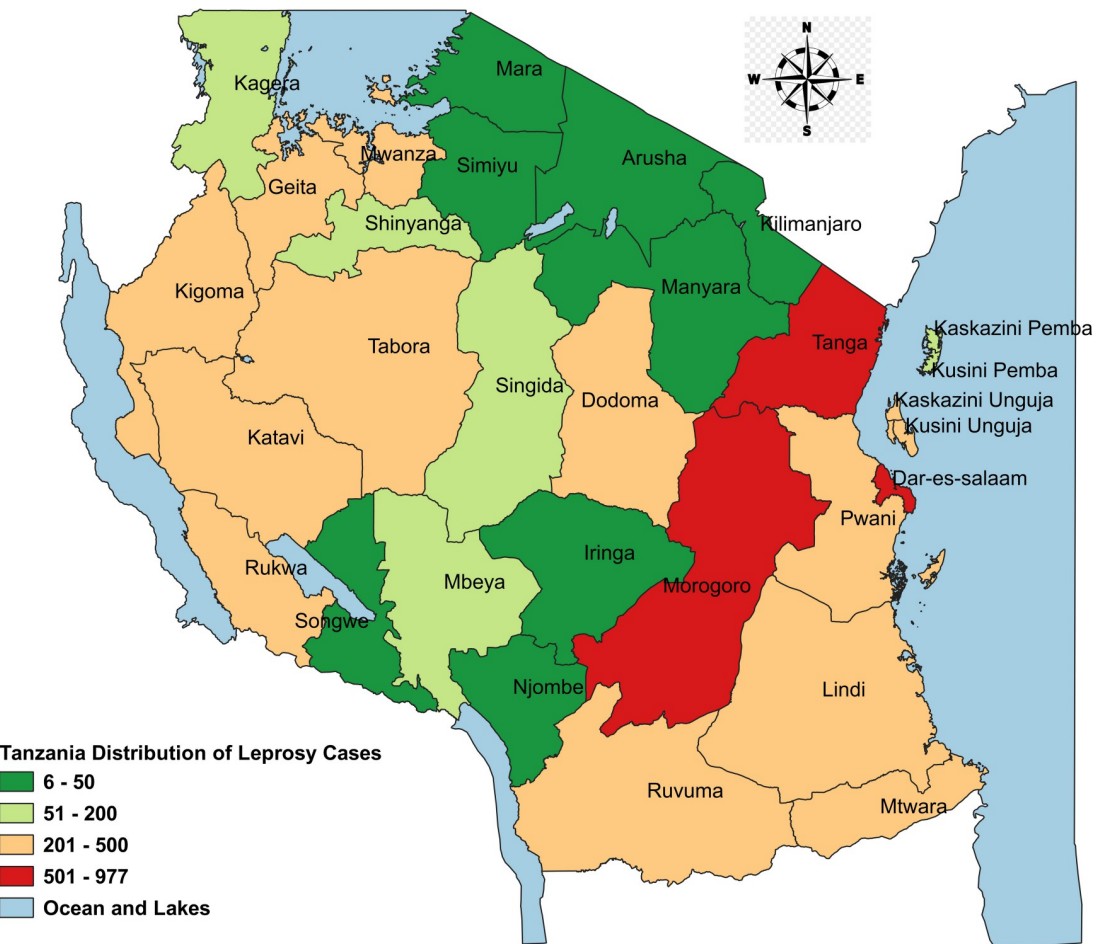

**Fig 1. Regional distribution of leprosy cases from 2017 to 2020, Tanzania.** Map was drawn using QGIS desktop software 3.26.3. The four-color coded regions show leprosy cases distribution from 2017–2020. The shapefiles used were from an openly available source (https://www.nbs.go.tz/statistics/topic/gis). The shapefiles were made based on the 2012 population and housing census, but in this study, shapefiles have been modified to capture all the regions information.

Moreover, there was a slight increase in newly reported cases of MB leprosy, rising from 92% to 93.3%, though it did not reach statistical significance (P = 0.083). The proportion of children diagnosed with leprosy remained stable at around 4% throughout the four years (P = 0.273) and the female proportion among new cases showed a slight decline from 36.7% to 34.8% (P = 0.299), though without statistical significance.

Lastly, the percentage of newly diagnosed cases with grade two disabilities saw a slight increase from 8.4% to 9.1%, but it was not statistically significant (P = 0.102). Table 3 provide detailed information.

From 2017 to 2020, eight (8) regions had a more than 50% decrease in the number of leprosy cases reported, with the highest reduction observed in Manyara, Mbeya and Songwe regions with a 100% (P = 0.124), 82.6% (P = 0.114) and 75% (P = 0.685) decrease respectively. However, the observed reduction in number of cases were not found to be statistically significant.

On the other hand, despite four (4) regions reporting an increase in number of leprosy cases for more than 50% between 2017 and 2020, once again, these observed increases were not found to be statistically significant. These regions were Katavi, Mara, Morogoro and

**Table 3. Trend in leprosy elimination indicators in Tanzania from 2017 to 2020.**

| Year | Population Estimate | Registered cases at end of year | New cases detected | Point prevalence per 10,000 population | NCDR per 100,000 population | % MB in new Cases | % Children in new cases | % Female in new cases | % G2D in new cases | G2D rate per 100,000 population |
|------|---------------------|---------------------------------|--------------------|----------------------------------------|------------------------------|-------------------|-------------------------|----------------------|---------------------|----------------------------------|
| 2017 | 51,557,363 | 1648 | 1585 | 0.32 | 3.1 | 92 | 4.1 | 36.7 | 8.4 | 0.28 |
| 2018 | 54,199,163 | 1924 | 1869 | 0.35 | 3.4 | 92.4 | 4.0 | 36.8 | 8.9 | 0.33 |
| 2019 | 55,890,747 | 1929 | 1823 | 0.35 | 3.3 | 93.2 | 3.9 | 34.0 | 9.1 | 0.35 |
| 2020 | 57,637,628 | 1462 | 1384 | 0.25 | 2.4 | 93.3 | 4.0 | 34.8 | 9.1 | 0.24 |
| | Trend test p value | | | P = 0.484 | P = 0.488 | P = 0.083 | P = 0.273 | P = 0.299 | P = 0.102 | P = 0.729 |

Unguja with 275% (P = 0.275), 225% (P = 0.197), 67.5% (P = 0.302) and 53.2% (P = 0.195) increase respectively. Moreover, there was no significant decrease in number of regions reporting more than 100 new cases per year from 7 regions in 2017 to 3 regions in 2020 (P = 0.180).

Although not statistically significant, majority of the regions have achieved a point prevalence of less than 1 per 10,000 population (P = 0.815). As for the year 2020, Unguja Island is the only region with a point prevalence of more than 1 per 10,000 population.

Unguja Island has routinely recorded a high rate of newly leprosy cases among children, ranging from 17% in 2017 to 20% in 2020 (P = 0.594). However, this difference in incidence among children was not statistically significant. Furthermore, there is no statistical significance increase in the number of regions reporting more than 20% of new leprosy cases with disability grade II, from 5 regions in 2017 to 7 regions in 2020 (P = 0.920).

Using LBS, number of regions with a high leprosy burden has remained relatively constant between 2017 and 2020, whereas 12 regions are highly burdened namely Tanga, Morogoro, Dar es Salaam, Unguja, Tabora, Shinyanga, Lindi, Pemba, Iringa, Rukwa, Mwanza and Kagera. S1 Table, S1 and S2 Figs provides detailed information.

## Factors associated with leprosy disability in Tanzania from the year 2017 to 2020

From January 2017 to December 2020, 6963 records were found in the NTLP database. Among these, 1,391 (20%) were omitted from the analysis of factors associated with disability because they missed disability grading and HIV status. The final analysis included 5,572 (80%) patients (Fig 2).

Information on factors linked to disability is provided in **Table 4.** The multiple logistic regression analysis confirmed the associations between five of the variables related to disability in the bivariate analysis. It was observed that male patients were 1.38 more likely to develop disability than female patients, AOR 1.38 95% CI (1.22–1.57). Patients aged 15 years and above, AOR 2.42 95% CI (1.60–3.67) were more likely to develop disability compared to those aged 14 years and below. Moreover, previously treated patients were twice as much more likely to develop disability than new patients, AOR 2.18 95% CI (1.69–2.82). Patients with positive HIV status were 1.6 times more likely to have disability than those with a negative HIV status, AOR 1.60 95% CI (1.11–2.30).

## Discussion

The findings of this study identify key risk factors for developing disabilities among leprosy patients which include male sex, older age, positive HIV status, and a history of previous treatment. No statistically significant differences were observed in point prevalence and new case detection rates (NCDR) throughout the study period.

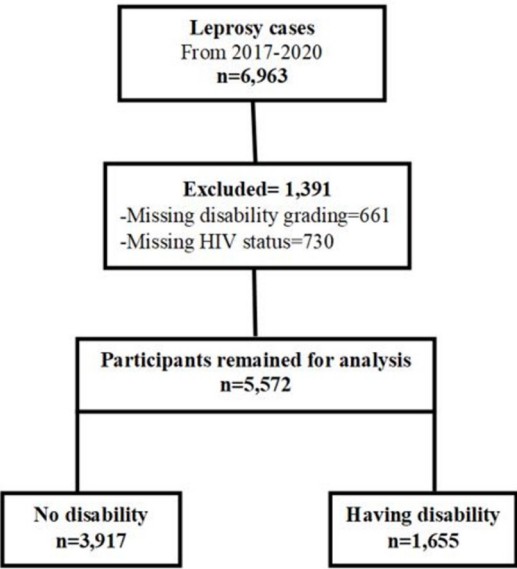

**Fig 2. Chart outlining how cases were included into the analysis of factors associated with disability among leprosy cases in Tanzania from the year 2017 to 2020.**

A majority of reported cases were male, accounting for 64.3% of total cases, resulting in a gender ratio of 1.8:1. These findings align with studies conducted in Africa, where the prevalence of leprosy in males ranged between 62% to 64.4% [14–16]. Similarly, studies in Asia also echoed this trend, revealing a predominant male presence in leprosy cases, ranging from 50.7% to 63.1% [17–19]. Potential reasons for higher leprosy occurrence in males in our local context may include social and cultural factors, where traditional gender roles expose men to specific transmission risks. Additionally, occupational factors play a significant role, as many cases are found in regions near lakes and oceans, where patients often work as fishermen,

**Table 4. Factors associated with disability among leprosy patients registered from January 2017 to December 2020, Tanzania.**

| Characteristics | Total | Having disability | Bivariate analysis | | Multivariate analysis | |
|---|---|---|---|---|---|---|
| | | | COR (95% CI) | P-value | Adjusted OR (95% CI) | P-value |
| **Gender** | | | | | | |
| Female | 1,989 | 504 (25.3) | **Reference** | | **Reference** | |
| Male | 3,583 | 1,151 (32.1) | 1.39 (1.23–1.58) | <0.001 | 1.38 (1.22–1.57) | <0.001 |
| **Age group (Years)** | | | | | | |
| ≤14 | 189 | 27 (14.3) | **Reference** | | **Reference** | |
| ≥15 | 5,383 | 1,628 (30.2) | 2.60 (1.72–3.93) | <0.001 | 2.42 (1.60–3.67) | <0.001 |
| **Category of Patient** | | | | | | |
| New | 5,321 | 1,537 (28.9) | **Reference** | | **Reference** | |
| Previous treated | 251 | 118 (47.0) | 2.18 (1.69–2.82) | <0.001 | 2.18 (1.69–2.82) | <0.001 |
| **Classification** | | | | | | |
| PB | 342 | 77 (22.5) | **Reference** | | **Reference** | |
| MB | 5,230 | 1,578 (30.2) | 1.49 (1.15–1.93) | 0.003 | 1.29 (0.99–1.68) | 0.059 |
| **HIV Status** | | | | | | |
| Negative | 5,444 | 1,604 (29.5) | **Reference** | | **Reference** | |
| Positive | 128 | 51 (39.8) | 1.59 (1.11–2.27) | 0.012 | 1.60 (1.11–2.30) | 0.011 |

leading to prolonged contact with potential carriers. Besides, majority of reported leprosy cases were above 15 years old, a trend that aligns with findings from studies conducted in both Africa and Asia [14–19]. Furthermore, significant proportion of the reported cases were categorized as MB. These results were consistent with previous studies conducted in Africa and Asia, which found that the majority of leprosy cases were classified as MB, with rates varying from 65% to 90.9% [15, 16, 18, 19]. High proportion of MB cases reflects ongoing disease transmission, given that MB patients are more contagious.

In this study, it was observed that there was a reduction in NCDR by 22.6% when comparing the year 2017 and 2020. It is worth noting that this reduction rate appears lower when compared to other studies, where substantial reductions ranging from 56.4% to 85.3% were reported [14, 18, 20, 21]. Similarly, the study recorded 21.9% reduction in point prevalence, however the observed reduction falls considerably below the impressive 78% reduction reported in a study conducted in Cameroon [20]. However, between 2017 and 2020, there was a rise in both NCDR and point prevalence. This increase can be attributed to fluctuations in funding availability for supporting active case findings. The country often depends on partner support for this, or on funds allocated in health facility and Comprehensive Council Health Plans which is not always consistent. It is evident from these findings that there is a need to enhance active case findings in order to boost leprosy case detection in the country. This is consistent with other studies and reports that have demonstrated active case finding as an effective strategy for enhancing case detection [22–24]. The percentage of females among new leprosy patients in our study fell between 34% to 36.7%, slightly below the recommended target of 40% and above. This discovery highlights the need for additional efforts to identify more female cases, as overlooking them raises the risk of ongoing transmission within households. The average proportion of new patient with grade II disability was found to be 8.9% during the 4 years period which was within the recommended threshold of less than 10%. The observed proportion was lower than other studies where proportion of grade II disability were recorded to be ranging between 19% and 45.8% [14, 18, 21]. The lower proportion of patients with leprosy disability grade II that was observed might be attributed to the continuous efforts made by the government to enhance the accessibility of leprosy services, which involves the timely detection and initiation of multidrug therapy.

Findings from our study revealed that there was a strong association between the age of the patient and disability where it was observed that the patients aged 15 years and above had higher risk of having disability. These results were consistent with other studies which affirmed the relationship between age and disability where it was noted that as age increases, the risk also increases [25–30]. Possible reasons for increased risk with age could be due to the fact that older individuals face challenges in recognizing early symptoms or seeking medical attention promptly and also older individual experience a longer exposure to risk factors.

Several studies have reported an association between gender and disability risk factors, with males being the most affected [25–27]. Our study's findings were consistent with these observations, as we also found that males were more likely to have a disability compared to females. This higher incidence among men may be attributed to specific behavior patterns influenced by social, cultural, and economic factors. Additionally, men tend to exhibit different patterns of health-care seeking behavior compared to women [28].

Similarly, patients who were previously treated were strongly associated with disability compare to newly diagnosed patients. The observed findings was similar to study done in Ethiopia which reported patient who defaulted treatment to have 15 times increased risk of disability compared to new patients [31]. Possible reasons for the increased risk of physical disability in leprosy patients with a history of previous treatment could be the development of drug

resistance. This resistance can lead to disease progression, resulting in complications, including nerve damage, and ultimately causing disability.

In the current study it was observed that, HIV positive clients were at increased risk of disability. This could be due to impaired immunity as a result of HIV disease causing leprosy bacteria to multiply more rapidly and cause more extensive nerve damage, leading to physical disability. Additionally, delay in diagnosis can occur in these patients since HIV-positive individuals may have atypical clinical presentations of leprosy, making diagnosis more challenging. Moreover, both HIV and leprosy can affect the nervous system increasing the severity and speed of nerve impairment, resulting in physical disabilities [32–34].

While our study contributes valuable insights into the burden of leprosy and associated risk factors related to disabilities, it is essential to acknowledge certain limitations that may influence the interpretation of our findings. The chronic nature of leprosy, coupled with its prolonged incubation period, poses a challenge in comprehensively capturing the disease's pattern within the confines of a four-year timeframe. Additionally, reliance on secondary data restricted our analysis to variables available in the database, preventing the assessment of other risk factors such as occupation, education level, patient delay, and nerve lesions.

## Conclusion

This study revealed that male sex, older patients, positive HIV status and having history of previous treatment to be independent risk factors for developing disability among leprosy patients. Additionally, there were no statistically significant differences in point prevalence and new case detection rate during the study period. To address leprosy-related disabilities, it is crucial to implement specific prevention strategies that focus on high-risk groups. This can be achieved by enhancing screening and contact tracing activities for the early identification of patients, thereby preventing delays in intervention. Additionally, further research is essential to analyze the long-term burden of leprosy in the country and identify any potential risk factors that were not explored in this particular study.

## Supporting information

**S1 Table. Trend in leprosy elimination indicators by regions from the year 2017 to 2020.** (PDF)

**S1 Fig. Distribution of leprosy burden in Tanzania regions for the year 2017.** Map was drawn using QGIS desktop software 3.26.3. The shapefiles used were from an openly available source (https://www.nbs.go.tz/statistics/topic/gis) The shapefiles were made based on the 2012 population and housing census, but in this study, shapefiles have been modified to capture all the regions information.
(TIF)

**S2 Fig. Distribution of leprosy burden in Tanzania regions for the year 2020.** Map was drawn using QGIS desktop software 3.26.3. The shapefiles used were from an openly available source (https://www.nbs.go.tz/statistics/topic/gis). The shapefiles were made based on the 2012 population and housing census, but in this study, shapefiles have been modified to capture all the regions information.
(TIF)

**S1 Data.**
(XLSX)

## Acknowledgments

The completion of this study was successful through joint efforts by members from Muhimbili University of Health and Allied Science, Ministry of Health, Tanzania Field Epidemiology and Laboratory Training Program (TFELTP), Centre for Disease Control and Prevention (CDC) through Mzumbe University and different individuals.

## Author Contributions

**Conceptualization:** George Mrema, Ally Hussein, Welema Magoge, Vida Mmbaga, Gideon Kwesigabo.

**Data curation:** George Mrema.

**Formal analysis:** George Mrema.

**Investigation:** George Mrema, Ally Hussein, Welema Magoge, Vida Mmbaga, Azma Simba, Emmanuel Nkiligi, Deus Kamara, Gideon Kwesigabo.

**Methodology:** George Mrema, Ally Hussein, Welema Magoge, Gideon Kwesigabo.

**Software:** George Mrema.

**Supervision:** Ally Hussein, Welema Magoge, Vida Mmbaga, Azma Simba, Robert Balama, Emmanuel Nkiligi, Paul Shunda, Deus Kamara, Riziki Kisonga, Gideon Kwesigabo.

**Validation:** George Mrema, Ally Hussein, Welema Magoge, Azma Simba, Robert Balama, Emmanuel Nkiligi, Paul Shunda, Riziki Kisonga, Gideon Kwesigabo.

**Visualization:** George Mrema, Ally Hussein, Welema Magoge, Vida Mmbaga, Azma Simba, Robert Balama, Emmanuel Nkiligi, Paul Shunda, Deus Kamara, Riziki Kisonga, Gideon Kwesigabo.

**Writing – original draft:** George Mrema, Ally Hussein, Gideon Kwesigabo.

**Writing – review & editing:** George Mrema, Ally Hussein, Welema Magoge, Vida Mmbaga, Azma Simba, Robert Balama, Emmanuel Nkiligi, Paul Shunda, Deus Kamara, Riziki Kisonga, Gideon Kwesigabo.

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
