## [Decision Letter · Decision Letter 0]

20 Nov 2023

PONE-D-23-30254Burden of Leprosy and associated risk factors for disabilities in Tanzania from 2017 to 2020: Analytical cross section studyPLOS ONE

Dear Dr. Mrema,

Thank you for submitting your manuscript to PLOS ONE. After careful consideration, we feel that it has merit but does not fully meet PLOS ONE’s publication criteria as it currently stands. Therefore, we invite you to submit a revised version of the manuscript that addresses the points raised during the review process.

Dear Dr. Mrema,

We have received valuable comments and feedback on your manuscript from both reviewers 1 and 2. Most comments indicated substantial revision before it is considered for publication while some of the comments needed clarifications. Most importantly, reviewers indicated that a major portion of conclusions and recommendations made were not supported by findings. Address all the comments provided by reviewers and editors. 

Editor’s comment

•             The finding mentioned in the abstract section Line #21 “point prevalence new case detection rate remained the same over the study period.” and the one mentioned in lines #164-165 “Over the years 2017 to 2020, the country observed a decline in the point prevalence of Leprosy 165 from 0.32 to 0.25 per 10,000 people” is not consistent.

•             Study design: instead of “analytical cross-sectional study’ I suggest ‘A retrospective cross-sectional study’

•             I suggest including a map that shows the distribution of leprosy in the country.

•             The first paragraph in the discussion section looks like a conclusion you may move it to the conclusion section. I suggest removing the subheading within the discussion.

•             Finding related to point prevalence (burden) is not sufficiently discussed. 

•             Provided that leprosy is a chronic disease with a long incubation period; is it possible to determine the pattern of the disease within 4 years?  The conclusion made may be misleading “No significant change in point prevalence and new case detection rate in the country was observed over the past four years”

We look forward to receiving your revised manuscript.

Kind regards,

Musa Mohammed Ali, PhD

Academic Editor

PLOS ONE

6. Please upload a copy of Figure 1 and 2, to which you refer in your text on page 16. If the figure is no longer to be included as part of the submission please remove all reference to it within the text.

7. Please include a copy of Table 4 which you refer to in your text on page 16.

8. We note that Figure S1 and S2 in your submission contain [map/satellite] images which may be copyrighted. All PLOS content is published under the Creative Commons Attribution License (CC BY 4.0), which means that the manuscript, images, and Supporting Information files will be freely available online, and any third party is permitted to access, download, copy, distribute, and use these materials in any way, even commercially, with proper attribution. For these reasons, we cannot publish previously copyrighted maps or satellite images created using proprietary data, such as Google software (Google Maps, Street View, and Earth). For more information, see our copyright guidelines: http://journals.plos.org/plosone/s/licenses-and-copyright.

a. You may seek permission from the original copyright holder of Figure S1 and S2 to publish the content specifically under the CC BY 4.0 license.  

Reviewers' comments:

Reviewer's Responses to Questions

**Comments to the Author**

1. Is the manuscript technically sound, and do the data support the conclusions?

Reviewer #1: Partly

Reviewer #2: No

2. Has the statistical analysis been performed appropriately and rigorously? 

Reviewer #1: Yes

Reviewer #2: No

3. Have the authors made all data underlying the findings in their manuscript fully available?

Reviewer #1: Yes

Reviewer #2: No

4. Is the manuscript presented in an intelligible fashion and written in standard English?

Reviewer #1: Yes

Reviewer #2: Yes

5. Review Comments to the Author

Reviewer #1: General remarks.

The authors have studied the trends in leprosy burden in Tanzania between 2017 and 2020 and factors associated with disability in leprosy patients. The findings have shown a very slight and insignificant reduction in leprosy burden in terms of prevalence, new case detection and grade-2-disabilities. Factors found to be associated with disability have mainly been demographic (age, sex) and disease related (form of leprosy, patient category & co-infection with HIV) in nature. Methods of leprosy data capture have been described but no data on the effectiveness or not of programmatic interventions/strategies has been given. However, a major portion of the conclusion and recommendations are about development and reinforcement of leprosy interventions and strategies, which are not really supported by findings.

Requests / edits

Need for more information on leprosy elimination program activities effectiveness/ineffectiveness in Tanzania that will support recommendations made on the manuscript.

Table 2: Your data revealed that 64.3% of males compared to 35.7% had leprosy. Did you try to find out any reason for this in the Tanzanian setting? Is there any fundamental difference in sex making the male gender more likely to develop leprosy compared to the female gender? Could it be an equity issue where the female gender has lesser access to leprosy services compared to the male gender?

Table 5: The HIV category “Unknown” is 730 with an AOR of 1.93 likely to develop G2D compared to HIV negative. Could including unknown HIV status in the analysis not influence the outcome?

Lines 11, 34-35: mycobacterium leprae is a scientific name and thus should be written in italics with the “M” in capital,

Line 36: important to precise “nasal droplets” instead of droplets simply.

Line 40: Complications instead of complication

Line 47: it is important to precise whether the authors are talking about 1000 new leprosy cases on 1000 leprosy cases. These do not mean the same thing.

Lines 49-52: I think that if you are talking about data from a single year (2021 in this cases), you can not use trends to describe difference in new case detection among WHO regions.

Line 55. Were the 10 cases per 100,000 population reported by all the 14 councils put together or by each one of the 14 councils?

Lines 61-64: The first letter of word leprosy is capitalized within sentences. Is there any particular reason for this?

Line 92: dot is an abbreviation for “directly observed treatment” and should thus be written in capital and the full meaning given in bracket here where it has been used for the first time.

Line 102: its should be replaced with is.

Line 150: Do you mean the National TB and Leprosy Program or TB Program?

Line 159 and Table 2: There is a discrepancy in the proportion of G2D cases reported: 10.4% vs 9.4%

Lines 185-186: Reporting of increases in number of leprosy cases should be consistent with for all 4 regions mentioned. Use either percentage increase or point increase for all and not a mixture of percentage and point increase.

Lines 214-215: regarding age and disability, persons aged 15 and over (usually considered as adults in leprosy) were all likely to develop disability (AOR 2.32 – 2.79) compared to those 14 years of age and below (usually considered as children in leprosy). I think it would be better categorizing age into two categories: children (<=14 yrs) and adults (>=15yrs) and redo the analysis.

Reviewer #2: please see attached manuscript, minor changes regarding objective, methods under abstract to specify period, etc. In general this study is a secondary data review of national data..no specific interviews have been done on assessing the risk factors and just demographic factors seem to have been taken for logistic regression

6. PLOS authors have the option to publish the peer review history of their article (what does this mean?). If published, this will include your full peer review and any attached files.

Reviewer #1: No

Reviewer #2: **Yes: **SRINIVAS GOVINDARAJULU

---

## [Author Response · Author response to Decision Letter 0]

3 Jan 2024

We have taken into account all the comments and integrated their suggestions into the revised version of the manuscript. Additionally, we have attached response matrix highlighting how we have adressed all comments.

---

## [Decision Letter · Decision Letter 1]

4 Mar 2024

PONE-D-23-30254R1Burden of Leprosy and associated risk factors for disabilities in Tanzania from 2017 to 2020PLOS ONE

Dear Dr. Mrema,

Thank you for submitting your manuscript to PLOS ONE. After careful consideration, we feel that it has merit but does not fully meet PLOS ONE’s publication criteria as it currently stands. Therefore, we invite you to submit a revised version of the manuscript that addresses the points raised during the review process.

We look forward to receiving your revised manuscript.

Kind regards,

Musa Mohammed Ali, PhD

Academic Editor

PLOS ONE

Reviewers' comments:

Reviewer's Responses to Questions

**Comments to the Author**

1. If the authors have adequately addressed your comments raised in a previous round of review and you feel that this manuscript is now acceptable for publication, you may indicate that here to bypass the “Comments to the Author” section, enter your conflict of interest statement in the “Confidential to Editor” section, and submit your "Accept" recommendation.

Reviewer #1: (No Response)

Reviewer #2: (No Response)

2. Is the manuscript technically sound, and do the data support the conclusions?

Reviewer #1: Yes

Reviewer #2: No

3. Has the statistical analysis been performed appropriately and rigorously? 

Reviewer #1: Yes

Reviewer #2: No

4. Have the authors made all data underlying the findings in their manuscript fully available?

Reviewer #1: Yes

Reviewer #2: No

5. Is the manuscript presented in an intelligible fashion and written in standard English?

Reviewer #1: Yes

Reviewer #2: Yes

6. Review Comments to the Author

Reviewer #1: General remarks:

Most of the queries raised in the first review has been taken care of by the authors in Revision 1 of the manuscript.

Additional query:

I understand from information in lines 100 – 103, that leprosy detection in Tanzania is largely passive, with no major efforts by the National TB and Leprosy Control Programme to improve upon cases detection.

With this background, and with the graphical presentation of data (from Table 3 in the manuscript) below, could the authors make some clarifications on the increase of leprosy prevalence and case detection of 17% and 20% respectively from 2017 to 2018, followed by the remarkable decrease in by 24% between 2019 and 2020.

I feel that there are programmatic issues related to these trends that require elucidation.

See attachment.

Reviewer #2: With an objective to assess risk factor for disability, the study should have used primary data also and investigated the logical risk factors (lepra reactions, time delay from symptom noticed to diagnosis), etc.

7. PLOS authors have the option to publish the peer review history of their article (what does this mean?). If published, this will include your full peer review and any attached files.

Reviewer #1: **Yes: **Dr Earnest Njih Tabah, MD, MPH, PhD.

National Programme Manager, National Yaws, Leishmaniasis, Leprosy and Buruli Ulcer Control Programme, Ministry of Public Health, Yaounde-Cameroon.

Seniour Lecturer of Public Health and Epidemiology, Faculty of Medicine and Pharmaceutical Sciences, University of Dschang, West Region, Cameroon.

Reviewer #2: **Yes: **SRINIVAS GOVINDARAJULU

---

## [Author Response · Author response to Decision Letter 1]

19 Mar 2024

We appreciate all reviewers for their valuable comments, which have greatly enhanced the quality and content of our manuscript titled "Burden of Leprosy and associated risk factors for disabilities in Tanzania from 2017 to 2020". We have taken into account all the comments and integrated their suggestions into the revised version of the manuscript. We believe that we have effectively addressed the raised concerns and that, after incorporating the changes, our manuscript is now appropriate for publication.

---

## [Decision Letter · Decision Letter 2]

5 Jun 2024

PONE-D-23-30254R2Burden of Leprosy and associated risk factors for disabilities in Tanzania from 2017 to 2020PLOS ONE

Dear Dr. Mrema,

Thank you for submitting your manuscript to PLOS ONE. After careful consideration of the revised manuscript and reviewers review report, we feel that it has merit but does not fully meet PLOS ONE’s publication criteria as it currently stands. Therefore, we invite you to submit a revised version of the manuscript that addresses the points raised during the review process.

**Reviewer #3 has raised a specific issue that needs to be addressed. Before considering the manuscript for publication,  address the additional comments provided by reviewer #3?**

We look forward to receiving your revised manuscript.

Kind regards,

Musa Mohammed Ali, PhD

Academic Editor

PLOS ONE

Journal Requirements:

Reviewers' comments:

Reviewer's Responses to Questions

**Comments to the Author**

1. If the authors have adequately addressed your comments raised in a previous round of review and you feel that this manuscript is now acceptable for publication, you may indicate that here to bypass the “Comments to the Author” section, enter your conflict of interest statement in the “Confidential to Editor” section, and submit your "Accept" recommendation.

Reviewer #2: All comments have been addressed

Reviewer #3: All comments have been addressed

2. Is the manuscript technically sound, and do the data support the conclusions?

Reviewer #2: Yes

Reviewer #3: Yes

3. Has the statistical analysis been performed appropriately and rigorously? 

Reviewer #2: Yes

Reviewer #3: Yes

4. Have the authors made all data underlying the findings in their manuscript fully available?

Reviewer #2: Yes

Reviewer #3: Yes

5. Is the manuscript presented in an intelligible fashion and written in standard English?

Reviewer #2: Yes

Reviewer #3: Yes

6. Review Comments to the Author

**Reviewer #2: **The nature of data allows only this level of analysis; this may be mentioned as a limitation of the study

**Reviewer #3: **I still think some contextual and programmatic information are needed to explain the high leprosy burden 14 counties: either real high burden or because of availability of quality services, trained health care workers and less stigma in these counties compared to others. This information is necessary to support a stratified and targeted intervention as recommended by the author.

7. PLOS authors have the option to publish the peer review history of their article (what does this mean?). If published, this will include your full peer review and any attached files.

Reviewer #2: **Yes: **SRINIVAS GOVINDARAJULU

Reviewer #3: **Yes: **Gidado Mustapha

---

## [Author Response · Author response to Decision Letter 2]

21 Jun 2024

We thank the reviewer for the additional comments, which continue to improve the quality and content of our manuscript titled "Burden of Leprosy and associated risk factors for disabilities in Tanzania from 2017 to 2020". We have addressed the comments and integrated their suggestions into the revised version of the manuscript. We believe that we have effectively addressed the raised concerns and that, after incorporating the changes, our manuscript is now appropriate for publication.

---

## [Decision Letter · Decision Letter 3]

8 Aug 2024

PONE-D-23-30254R3Burden of Leprosy and associated risk factors for disabilities in Tanzania from 2017 to 2020PLOS ONE

Dear Dr. Mrema,

Thank you for submitting your manuscript to PLOS ONE. After careful consideration, we feel that it has merit but does not fully meet PLOS ONE’s publication criteria as it currently stands. Reviewer 3 suggested to edit the language. Therefore, we invite you to submit a revised version of the manuscript that addresses the points raised during the review process.

**ACADEMIC EDITOR: Reviewer suggested language editing  **

We look forward to receiving your revised manuscript.

Kind regards,

Musa Mohammed Ali, PhD

Academic Editor

PLOS ONE

Journal Requirements:

Reviewers' comments:

Reviewer's Responses to Questions

**Comments to the Author**

1. If the authors have adequately addressed your comments raised in a previous round of review and you feel that this manuscript is now acceptable for publication, you may indicate that here to bypass the “Comments to the Author” section, enter your conflict of interest statement in the “Confidential to Editor” section, and submit your "Accept" recommendation.

Reviewer #2: All comments have been addressed

Reviewer #3: All comments have been addressed

2. Is the manuscript technically sound, and do the data support the conclusions?

Reviewer #2: Partly

Reviewer #3: Yes

3. Has the statistical analysis been performed appropriately and rigorously? 

Reviewer #2: No

Reviewer #3: Yes

4. Have the authors made all data underlying the findings in their manuscript fully available?

Reviewer #2: Yes

Reviewer #3: Yes

5. Is the manuscript presented in an intelligible fashion and written in standard English?

Reviewer #2: Yes

Reviewer #3: Yes

6. Review Comments to the Author

Reviewer #2: Plan your future studies with a focused objective & select an appropriate design to study risk factors

Reviewer #3: Some level of English editing is still needed for the manuscript, the author should leverage the services of professional editors which can also help reduce redundancy in some of the sentences.

7. PLOS authors have the option to publish the peer review history of their article (what does this mean?). If published, this will include your full peer review and any attached files.

Reviewer #2: **Yes: **SRINIVAS GOVINDARAJULU

Reviewer #3: **Yes: **Gidado Mustapha

---

## [Author Response · Author response to Decision Letter 3]

24 Aug 2024

Dear Editor,

We are grateful to the reviewers for their additional comments, which further improve the quality of our manuscript titled "Burden of Leprosy and associated risk factors for disabilities in Tanzania from 2017 to 2020". We have taken the comments into account and incorporated their suggestions into the revised manuscript. We believe we have effectively addressed the concerns raised, and after making these changes, our manuscript is now suitable for publication.

---

## [Editor Report · Decision Letter 4]

24 Sep 2024

Burden of Leprosy and associated risk factors for disabilities in Tanzania from 2017 to 2020

PONE-D-23-30254R4

Dear Dr. Mrema,

We’re pleased to inform you that your manuscript has been judged scientifically suitable for publication and will be formally accepted for publication once it meets all outstanding technical requirements.

Kind regards,

Musa Mohammed Ali, PhD

Academic Editor

PLOS ONE
---

## [Editor Report · Acceptance letter]

29 Sep 2024

PONE-D-23-30254R4 

PLOS ONE

Dear Dr. Mrema, 

I'm pleased to inform you that your manuscript has been deemed suitable for publication in PLOS ONE. Congratulations! Your manuscript is now being handed over to our production team.

Kind regards, 

on behalf of

Dr. Musa Mohammed Ali 

Academic Editor

PLOS ONE